# Clinical Significance of Multiparameter Intracranial Pressure Monitoring in the Prognosis Prediction of Hypertensive Intracerebral Hemorrhage

**DOI:** 10.3390/jcm11030671

**Published:** 2022-01-28

**Authors:** Yongbo Yang, Yuchun Pan, Chunlei Chen, Penglai Zhao, Chunhua Hang

**Affiliations:** 1Department of Neurosurgery, The Affiliated Nanjing Drum Tower Hospital of Nanjing University Medical School, Nanjing 210008, China; Yangyongbo2000@163.com (Y.Y.); 15720801655@163.com (C.C.); 2Department of Neurosurgery, Nanjing Lishui People’s Hospital, Nanjing 211200, China; neurosurgeryp@163.com

**Keywords:** multi-parameter intracranial pressure monitoring, hypertensive intracerebral hemorrhage, brain injury

## Abstract

Objective: The present study aimed to investigate the clinical significance of multiparameter intracranial pressure (ICP) monitoring in the prediction of the prognosis of hypertensive intracerebral hemorrhage (HICH). Methods: A retrospective analysis was performed on the clinical data of 53 HICH patients. The patients underwent removal of intracranial hemorrhage and decompressive craniectomy after admission. A ventricular ICP monitoring probe was used to continuously and invasively monitor mean arterial pressure (MAP) and ICP after surgery. The NEUMATIC system was used to collect ICP data, including pressure reactivity index (PRx), ICP dose (DICP), amplitude and pressure regression (RAP), and cerebral perfusion pressure (CPP). The mean PRx, CPP, RAP, ICP, and DICP20 mmHg × h were calculated with 1 h as the time segment. According to the Glasgow outcome scale (GOS) scores after discharge, the patients were grouped into the poor prognosis group (GOS I–III) and the good prognosis group (GOS IV and V). The two groups were compared in terms of GOS scores in the treatment and prediction of prognosis of patients. Results: The good prognosis group showed significantly lower values of mean ICP, DICP20 mmHg × h, RAP, and PRx than the poor prognosis group, while CPP was significantly higher (*p* < 0.001). Conclusions: PRx, DICP, RAP, and CPP could reflect intracranial changes in patients and were significantly correlated with the prognosis of the patients. Mean ICP, PRx, DICP20 mmHg × h, and RAP were negatively correlated with prognosis, while CPP was positively correlated with prognosis.

## 1. Introduction

Patients with hypertensive intracerebral hemorrhage (HICH) exhibit pathological changes such as fibrous or hyalinized degeneration in the vascular wall of the intracranial arteriole due to lasting hypertension, which weakens the elasticity of the vascular wall, eventually leading to vascular rupture and bleeding. It is one of the most severe complications of hypertension, with 9–28% morbidity in Europe and the United States, and 19–48% in China, among patients with cerebral stroke [1,2,3]. HICH deteriorates rapidly and damages neurological function in the early stage, leading to high rates of mortality, disability, and morbidity [4,5,6,7,8]. It is also likely to recur, making it one of the major diseases that endangers human health. This indicates that high intracranial pressure (ICP) should be closely monitored during the treatment of patients with severe HICH. Furthermore, the frequency of intracranial hypertension is independently correlated with the mortality and prognosis of HICH, and all patients with HICH should undergo ICP monitoring (grade-B evidence for grade I) [9]. Consequently, HICH patients score 3–8 points on the Glasgow coma scale (GCS). Invasive ICP monitoring revealed grade-C (evidence for grade IIB), and all patients with ICP > 20 mmHg for a prolonged period should be treated with ICP lowering measures. Previous studies showed that the morbidity of intracranial hypertension in HICH patients was 67%, and the fatality rate of intracranial hypertension was 50% [5,6]. ICP monitoring has been widely applied in the clinical treatment of HICH. However, understanding of this phenomenon is limited to only mean ICP since data analysis methods are insufficient, and multiparameter analysis is based on severe traumatic brain injury (TBI) [10,11]. PRx, RAP, and CPP are hemodynamic indicators of intracranial vascular compliance, intracranial compensatory capacity, and cerebral blood supply that predict intracranial deterioration of the disease in patients and suggest the outcomes of the disease. DICP20 can reflect the fluctuation and duration of ICP over a period of time. Comprehensively, they can infer the cerebrovascular-related situation of patients and help doctors to judge the prognosis more scientifically. Therefore, it is critical to study multiple parameters, including the pressure reactivity index (PRx), the ICP dose (DICP), the regression of amplitude and pressure (RAP), and the cerebral perfusion pressure (CPP), of ICP monitoring in HICH cases to determine the scientific and theoretical basis for HICH and explore an optimal analysis approach to guide the prediction of prognosis. In this document, a retrospective analysis was performed on 53 HICH patients treated in the Department of Neurosurgery of Nanjing Lishui People’s Hospital, Lishui Region of the Zhongda Hospital, affiliated with Southeast University, Nanjing, China, from March 2018 to 2020. The ICP monitoring parameters of these patients were analyzed to investigate the importance of combined multiparameter monitoring analysis in the prediction of the prognosis of HICH.

## 2. Data and Methods

### 2.1. Patient Information and Grouping

A total of 53 HICH patients, including 27 males and 26 females, treated at our hospital were included in this study. All patients were hypertensive. In the emergency room, intravenous antihypertensive drugs have been used to control blood pressure. During the treatment in our department, the blood pressure of the patients was roughly controlled within the normal range. The GCS scores of these patients were between 4 and 9 points. The shape of the hematomas was oval, and all the hematomas were located in the basal ganglia, which did not show obvious compression on the brain stem. All volumes were between 20 and 50 mL. All patients’ families signed the informed consent for operation and ICP probe placement. The inclusion criteria were as follows: all patients had indications of emergency surgery and had undergone intracranial hematoma removal and craniotomy decompression (standard large trauma craniotomy), and a ventricular ICP probe was placed after admission. These patients were treated with ICP monitoring for 3 consecutive days after the operation. The exclusion criteria were as follows: patients with any underlying disease that would affect prognosis, including other cardiovascular and cerebrovascular diseases, liver or kidney dysfunction, and coagulation disorders or intracranial infection. According to the Glasgow outcome scale (GOS) scores at 6 months after discharge, patients were grouped into the poor prognosis group (GOS I–III) and the good prognosis group (GOS IV and V).

We used mannitol (125 mL) every 8 h when ICP was above 20 mmHg, and we also used mannitol (125 mL) every 6 h if this did not work. If ICP was greater than 25 mmHg after surgery, we usually treated it with mannitol (125 mL) every 6 h and furosemide (20 mg) once a day, or albumin (20 g) twice a day. To deal with the sudden and sharp increase in ICP, in addition to changing body position, strengthening nursing, and using drugs to reduce ICP, a timely review of head CT was essential to clarify the situation of intracranial hematoma to ensure whether operation was necessary again.

There were no new cerebral hemorrhages in all patients. In the poor prognosis group, one patient died, and one was in a vegetative state, but there was no death or vegetative state in the good prognosis group.

### 2.2. Data Extraction

ICP probes (Johnson & Johnson, New Brunswick, NJ, USA) were implanted in all patients using appropriate surgical procedures, then ICP probes were removed after 3 days. ICP-related data were collected using the NEUMATIC system (Shanghai Haoju Medical Technology Co., Ltd., Shanghai, China) and transmitted to a server for storage in real time. The measurement intervals for all data were 3 seconds and then were calculated with 1 h as the time segment. These parameters included ICP, PRx, RAP, CPP, and DICP obtained by calculating the area under the curve (AUC) that exceeded a threshold in the ICP curve. As the current recommended ICP threshold for HICH treatment is 20 mmHg, DICP20 mmHg × h was used in this study.

### 2.3. Statistical Analysis

SPSS 22.0 was used for statistical analysis, and GraphPad Prism 8.0 was used for image processing. Student’s t-test was used for intergroup comparisons of measurement data conforming to the normal distribution, and data were expressed as mean ± standard deviation. The Mann–Whitney U test was used for intergroup comparisons of measurement data not conforming to the normal distribution, and data were expressed as median and interquartile range. A chi-square test was used to compare the enumeration data expressed as the number of patients. A *p*-value < 0.05 indicated statistical significance. The receiver operating characteristic (ROC) curve was constructed, and the AUC was calculated to evaluate the significance of the parameters in predicting the prognosis of patients.

## 3. Results

### 3.1. Comparison of Baseline Information between the Two Groups

Parameters such as age, sex, GCS scores on admission, and hematoma volume did not differ significantly between the good prognosis group (*n* = 27) and the poor prognosis group (*n* = 26) (*p* > 0.05; Table 1). However, the good prognosis group showed lower mean ICP, RAP, DICP20 mmHg × h, and PRx, and higher CPP than the poor prognosis group (*p* < 0.001; Table 1).

### 3.2. Comparison of DICP20 mmHg × h Data between the Two Groups

The DICP20 mmHg × h data, which did not correspond to the normal distribution, were expressed as medians and quartiles, and indicated a significantly greater data dispersion of the poor prognosis group than of the good prognosis group (Figure 1).

### 3.3. Comparison of ROC Curves of the Significance of Different Parameters in Predicting a Poor Prognosis in Patients

The significance of the ICP parameters in the prognosis of HICH was analyzed. As shown in the ROC curves (Figure 2), the AUCs of the mean PRx, DICP20 mmHg × h, CPP, RAP, and ICP were 0.987 (95% confidence interval (CI): 0.965–1.000), 0.818 (95% CI: 0.703–0.932), 0.860 (95% CI: 0.746–0.974), 0.941 (95% CI: 0.881–1.000), and 0.967 (95% CI: 0.927–1.000), respectively. Mean PRx, CPP, RAP, ICP, and DICP20 mmHg × h, especially PRx, were very significant in predicting a poor prognosis in patients.

## 4. Discussion

Approximately two million patients die every year in China due to cerebrovascular diseases, making China the highest-ranking country in cerebrovascular disease mortality. Stroke was the most fatal among 25 diseases in China from 1997 to 2017, with a high disability rate and mortality [8]. The most common subtype under spontaneous intracerebral hemorrhage is HICH. The HICH-induced space-occupying effect might result in intracranial hypertension or even compressive displacement of remote brain tissues, causing numerous natural killer cells to infiltrate brain tissues around the hematoma due to immune screening, thus releasing cytotoxic molecules that directly damage the blood–brain barrier and recruit other immune cells to exacerbate the inflammatory response [7]. Therefore, the primary objective of surgical treatment for HICH is to reduce ICP, and ICP management during the perioperative period is one of the key therapies for HICH [8]. In addition to high ICP, the automatic regulation disorder of intracranial vessels and hemodynamic changes caused by hypoxia of brain tissue and the generation of free radicals also affect the prognosis of HICH. Therefore, not only should the mean changes in ICP caused by the space-occupying effect, but also the hemodynamic indicators reflecting the pathological changes of the cerebral vessels, be closely monitored during the treatment of HICH. PRx, RAP, and CPP could be applied to study the prognosis prediction of HICH prognosis given their importance in the prediction of TBI prognosis [8,10,11].

Lundberg et al. developed modern ICP technology for direct measurement, which is an essential and widely used method to monitor ICP of patients with severe neurological diseases in clinical practice [12,13]. The mean ICP is critical in guiding the timing of surgery and standardized doses of dehydration drugs, which could improve the prognosis of patients and reduce the duration of hospitalization [14,15]. In clinical application, the ICP threshold guides the treatment of diseases. However, it is not the only observational indicator of treatment. Treatment should be based on a comprehensive analysis of several parameters related to ICP [16,17]. Since the mean ICP is not adequately efficient, the theoretical indicator DICP that reflects the ICP of a patient over a period was proposed. DICP is obtained by calculating the AUC that exceeds a threshold in the ICP curve, which can show the relationship between the extent to which the threshold is exceeded and the amount of time that the threshold is exceeded, to a certain extent. It has been shown to indicate the duration and degree of intracranial injury and predict prognosis better than the mean ICP [18,19]. According to the results of the present study, the mean ICP, DICP20 mmHg × h, and the data dispersion of DICP20 mmHg × h of the good prognosis group were significantly lower than those of the poor prognosis group. According to the ROC curves, both DICP20 mmHg × h and the mean ICP were critical for the prediction of the prognosis, although the former was poorer than the latter. These results were contradictory to those of Vik et al. [20]. However, selection bias as a result of the limited sample size needs to be addressed with a large sample size. Wu et al. [8] demonstrated that in TBI treatment, the mean ICP was an independent risk factor affecting the prognosis of the patients, while the initial ICP pressure predicted the prognosis of the patients better than the mean ICP. During the study, analysis of the very few clinical data revealed that among patients with similar DICP20 mmHg × h, most of them showed a higher ICP in the short term and had a better prognosis than those with slightly higher ICP (higher than the threshold but lower than the former) in the long term as a result of the effect of the initial ICP, thus suggesting that both the degree and duration of the increase in ICP could affect the prognosis of patients. Therefore, analyzing both DICP20 mmHg × h and the mean ICP, as well as the importance of the initial ICP in the prediction of the prognosis of HICH, is crucial.

In the pathogenesis of HICH, the physiological dysfunction of cerebral vessels is also an important factor. PRx, RAP, and CPP are hemodynamic indicators of intracranial vascular compliance, intracranial compensatory capacity, and cerebral blood supply that predict intracranial deterioration of the disease in patients and suggest the results of the disease [21]. CPP can be derived by calculating the difference between mean arterial pressure (MAP) and ICP, and it reflects cerebral blood flow (CBF). Thus, CPP values indirectly reflect the nutritional status of brain tissue, as well as the response of cerebrovascular autoregulation to fluctuations in blood pressure. The development of ICP-guided therapy to CPP-guided therapy raised increasing concern with regards to cerebral hemodynamics in the treatment process. In the treatment of HICH, sufficient CPP should be guaranteed and specific CBF should be maintained to minimize the degree of secondary ischemic brain injury [22,23]. At normal cerebrovascular reactivity (CVR), the resisting cerebral vessels are dilated when MAP drops to increase ICP and maintain CBF. In the event of CVR impairment, the resisting vessels cannot be dilated accordingly, causing a decrease in CBF and indicating a reaction correlation between MAP and ICP [24,25,26]. Based on this theory, the PRx indicator established a correlation between MAP and ICP with a value range from −1 to 1. PRx is a correlation index between arterial blood pressure and ICP that is mainly used to dynamically evaluate the autonomous regulatory ability of cerebral blood, and PRx values can also indicate the severity of the disease. A negative value indicates unimpaired CVR, normal automatic regulation of cerebral vessels, and improved prognosis in patients. PRx has also been proven to be more reliable than the ICP threshold alone in predicting death [23,27,28]. The results of this study showed that the good prognosis group had significantly lower PRx than the poor prognosis group, while no consensus was reached on the threshold, and no conclusions were reached on the tolerable degree of damage to the automatic regulation of cerebral vessels in patients. The correlation between MAP and ICP suggested that CPP can determine CBF and the amount of cerebral oxygen supply. Neurological dysfunction and secondary brain injury in HICH patients are ascribed to reduced blood flow and microcirculatory perfusion in the brain, which further worsen brain edema and the disease in the short term as a vicious cycle. Therefore, CPP-guided clinical treatment has been widely promoted, as CPP fluctuations below a certain threshold indicate stable disease status and good prognosis. In the existing TBI treatment, the CPP threshold has been determined as 50–70 mmHg, while in the HICH treatment, it has not been unified and determined. The results of this study showed that the good prognosis group had a significantly higher mean CPP than the poor prognosis group. In terms of the CPP threshold, CPP-guided therapy has improved in recent years, suggesting that the optimal cerebral perfusion pressure (CPPopt) can be calculated based on the PRx–ICP curve. In addition, individualized treatment could be effective for automatic regulation of cerebral vessels, indicating the tolerable degree of damage to automatic regulation of cerebral vessels in patients. The CPPopt theory offers another approach to the treatment of HICH [29]. However, all the previous studies had a small sample size. Therefore, we proposed that the current theory is scientific, and CPPopt-guided therapy will be widely used in future treatment as some HICH patients with hypertension have high initial blood pressure. Additionally, their vessels encounter lasting high pressure, and the cerebral blood supply is dynamically balanced on the basis of high MAP. Thus, these patients might have insufficient CPP if the CPP threshold is similar to that of HICH patients without hypertension.

According to the Monro–Kellie doctrine for intracranial pressure, when the intracranial space remains unchanged, a small increase in intracranial volume may result in a significant increase in ICP when the cerebral compensatory mechanism is completely destroyed, thus resulting in a decrease in CPP and CBF, and secondary injury of cerebral tissues. Therefore, intracranial compensation is critical to guide the prognosis of HICH, as well as RAP, the correlation coefficient between ICP volatility and ICP, was proposed. RAP is the correlation coefficient between the amplitude of the ICP and the ICP, and the values reflect cerebrovascular compliance and the compensatory reserve capacity of the cerebrospinal fluid. According to the analysis of the ICP volume curve by Czosnyka and Pickard [30], the ICP compensation increased and volatility decreased when the cerebral compensatory ability was at the normal value, while the RAP was 0, indicating that altered intracranial volume did not affect the ICP. When the cerebral compensatory capacity decreased, both ICP and ICP volatility increased, while RAP was 1 and was located in the steep region on the right side of the ICP volume curve, indicating that ICP changed with intracranial volume. When the cerebral compensatory capacity was exhausted, ICP increased continuously, while ICP volatility decreased and RAP dropped to a negative value, indicating that ICP changed markedly in the short term due to a small increase in intracranial volume [31]. Therefore, RAP can reflect intracranial compensatory ability and CVR, thus guiding treatment and indicating the prognosis of patients.

ICP monitoring data suggested various pathological changes in patients. The conventional mean ICP reflects the change in ICP, suggesting the intracranial space-occupying effect, and guides dehydration and surgical treatment. DICP20 mmHg × h can reflect both the volatility and duration of the ICP change in patients. PRx, CPP, and RAP indicate hemodynamic changes, such as the regulation of cerebral vessels in patients, and CPP can guide treatment and predict prognosis. ROC curves showed that ICP, DICP20 mmHg × h, PRx, CPP, and RAP, especially PRx, can predict patient prognosis. However, the strengths and weaknesses of these parameters and their significance in the prediction need to be substantiated using a large sample size.

## 5. Conclusions

In summary, the mean ICP is not the only indicator in HICH treatment. DICP20 mmHg × h, PRx, CPP, and RAP should be used together with the mean ICP to analyze the disease, guide treatment, and predict prognosis. Among these, the mean ICP, PRx, DICP20 mmHg × h, and RAP were negatively correlated with patient prognosis, while CPP showed a positive correlation. ICP, DICP20 mmHg × h, PRx, CPP, and RAP were significant for prognostic prediction. Of these, PRx was pivotal, and CPPopt-guided individualized treatment may drive future approaches.

## Figures and Tables

**Figure 1 jcm-11-00671-f001:**
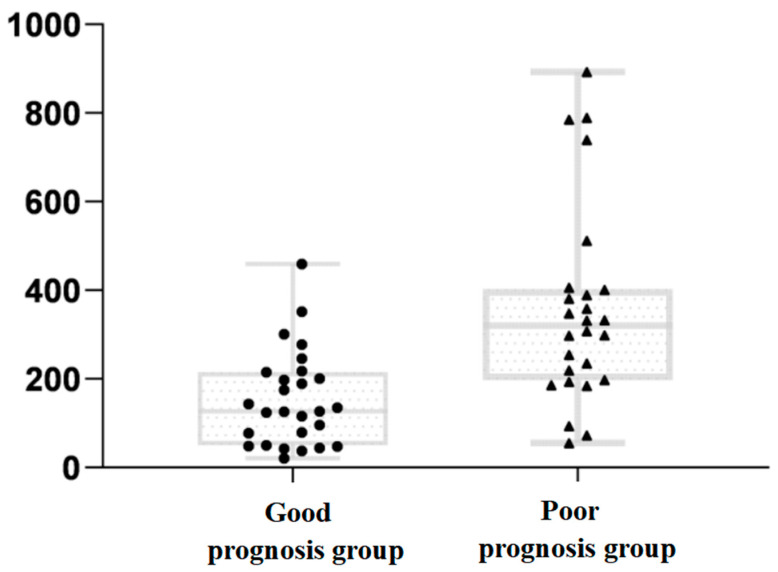
Comparison of detailed DICP20 mmHg × h data dispersion between the two groups. The square and triangle of pic in caption means the raw data.

**Figure 2 jcm-11-00671-f002:**
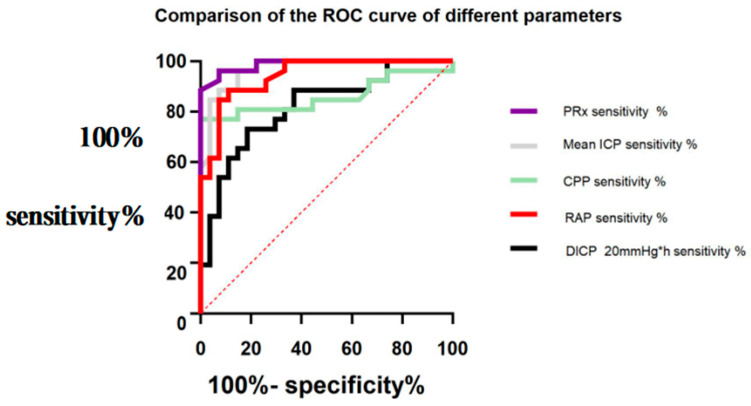
Comparison of the receiver operating characteristic (ROC) curves of the significance of different parameters in predicting the poor prognosis of patients.

**Table 1 jcm-11-00671-t001:** Comparison of baseline information between the two groups.

	Poor Prognosis Group	Good Prognosis Group	Current Result	*p*
Age (mean ± standard deviation, years)	64.58 ± 13.62	63.67 ± 11.84	t = 0.260	0.796
Gender (male/female, patient)	13:13	14:13	x^2^ = 0.018	0.893
GCS score on admission (M(IQR), points)	6 (7)	7 (7)	U = 326	0.650
Hematoma volume (mL)	37.38 ± 6.40	35.41 ± 6.63	t = 1.10	0.275
Mean ICP (mean ± standard deviation, mmHg)	27.68 ± 13.17	10.12 ± 4.37	t = 6.566	<0.001
PRx (mean ± standard deviation)	0.34 ± 0.12	0.05 ± 0.98	t = 9.869	<0.001
RAP (mean ± standard deviation)	0.37 ± 0.12	0.14 ± 0.10	t = 7.619	<0.001
CPP (mean ± standard deviation, mmHg)	55.88 ± 14.68	73.65 ± 8.27	t = 5.454	<0.001
DICP20 (M (IQR), mmHg × h)	320.36 (403.775)	127.25 (215.64)	U = 574.00	<0.001

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
