# Peer review of "Clinical Significance of Multiparameter Intracranial Pressure Monitoring in the Prognosis Prediction of Hypertensive Intracerebral Hemorrhage"

_jcm, 2022, doi:10.3390/jcm11030671_

Round 1

Reviewer 1 Report

In this retrospective study by Yang et al., they correlate various continuous parameters of intracranial pressure with prognostic outcome in a cohort of hypertensive intracerebral hemorrhage patients. Yang et al., report that following emergency decompressive surgery various acute time dose measurements of intracranial changes could hold prognostic value for these patients. Although derived from a relatively small cohort of patients, the reported values of continuous monitoring in these patients is interesting The statistical approach in manuscript analyses is sound and manuscript is well written.

Authors should address various issues to strengthen the manuscript:

  • Focus on the hypertensive ICH subgroup is pragmatic in the need for decompressive surgery and subsequent feasibility in the placement of measuring probe; do authors have corresponding data for hemorrhagic stroke caused by other factors, aneurysm/vascular malformation etc which similarly require craniectomy? Comparative differences would be interesting to see.
  • The heatmap in Fig. 2, is this data paired or unpaired? If paired, how did authors designate patient-patient comparison? Fig. 1 seems to be a better graphic representation of their data…
  • In the continuous measurement of both groups were trends detected in the various parameter changes after surgery or treatment, or between groups. These data should replace Fig. 2.

Minor Issues:

  • Include a glossary of abbreviation
  • Suboptimal data presentation in Fig. 3. Difficult to read, should use contrasting colors, increase resolution and size
  • Add citation #7 to line 145
  • Correct line 242

Author Response

Q1:The heatmap in Fig. 2, is this data paired or unpaired? If paired, how did authors designate patient-patient comparison? Fig. 1 seems to be a better graphic representation of their data…In the continuous measurement of both groups were trends detected in the various parameter changes after surgery or treatment, or between groups. These data should replace Fig. 2.

Response:Thank you for your suggestion. After careful analysis, we found that Figure 1 includes Figure 2, so we have deleted Figure 2. Sorry, this is our negligence in considering the problem. Please forgive me.

Q2:Include a glossary of abbreviation.

Response:Thank you for your advice. We have revised the manuscript and added the glossary of abbreviation in the end of the text.

Q3:Suboptimal data presentation in Fig. 3. Difficult to read, should use contrasting colors, increase resolution and size.

Response:This is our negligence. Thank you for your advice. We have redrawn Figure 3 (in fact the figure is Figure 2 now)according to your requirements.

Q4:Add citation #7 to line 145;Correct line 242.

Response:This is our negligence. Thank you for your advice.We have revised the manuscript.

Reviewer 2 Report

Interesting retrospective study describing brain pressure-related parameters in patients suffering from ICH.

Outcome groups should be named as poor and good.

In Table 1 GCS should be expressed as median.

Figure 3 is of poor quality - lines are too thin

References need attention- first names and family names are often mixed.

Lower RAP is usually in acute ilnesses associated with poor outcome- please check statistiocs

Author Response

Q1:Outcome groups should be named as poor and good.

Response:Thank you for your advice. This is an oversight of our work,we have made changes in the manuscript.

Q2:In Table 1 GCS should be expressed as median.

Response:Thank you for your suggestion. I'm deeply sorry for our statistical error,however, the statistical methods used in the analysis of GCS in some literatures are the same as ours. We have modified Table 1 according to your suggestion.

Q3:Figure 3 is of poor quality - lines are too thin

Response:This is our negligence. Thank you for your advice. We have redrawn Figure 3(in fact the figure is Figure 2 now)according to your requirements.

Q4:References need attention- first names and family names are often mixed.

Response:We apologize for this problem, we have reviewed the references according to your suggestions. Thank you for your guidance.

Q5:Lower RAP is usually in acute ilnesses associated with poor outcome- please check statistiocs.

Response:Thanks to the professor's suggestion, we reviewed the original data again, re analyzed the data, and the results were correct. At the same time, we searched the relevant literature about RAP, and the results pointed out that lower RAP is usually in acute ilnesses associated with good outcome.

Wang T,Liang RC,Feng JF,et al.Prognostic value of intracranial pressure related parameters in patients with traumatic brain injury [J].Chin J Neurosurg,2020,36(10):1021-1025.

Reviewer 3 Report

In this study, Yang et al. aim to determine whether physiological parameters other than mean intracranial pressure can improve prognostication of hypertensive intracerebral hemorrhage (HICH) patients at six months after the hemorrhage. In addition, they aim to determine whether these parameters can be used to inform treatment of HICH. To do this, they collect data on intracranial pressure (ICP) and mean arterial pressure in a cohort of 52 HICH patients which they then use to calculate additional parameters such as the pressure reactivity index (PRx), intracranial pressure dose (DICP), amplitude and pressure regression (RAP), and cerebral perfusion pressure (CPP). Patients are then grouped according to good or poor Glasgow outcome scale scores at six months post hemorrhage and the parameters compared between the two groups. They find that mean ICP, PRx, RAP, and DICP are significantly elevated in patients with poor prognosis while CPP is reduced in patients with a poor prognosis. An ROC analysis revealed that each of these parameters was capable of HICH prognosis at 6 months with PRx and mean ICP being the most accurate. The authors interpret that these results indicate that there is more than one parameter that informs HICH treatment.

While the study is valid there are several major flaws in this manuscript that must be rectified before I can recommend it for publication:

Major comments:

  • Data on hematoma volume (pre- and post-operative) and location needs to be included. Hematoma volume and location is a powerful prognostic and is a confounding variable that needs to be accounted for.
  • Data on baseline blood pressure not given therefor not possible to determine whether patients are hypertensive.
  • This study does not inform better treatment of HICH. Use of CPPopt in HICH has been previously reported (“Optimal cerebral perfusion pressure in patients with intracerebral hemorrhage: an observational case series” PMID: 24666981). Would advise that the authors adjust their manuscript to focus on prognostication.
  • References missing (e.g. Lines 45-47 refer to “Guidelines for the Management of Spontaneous Intracerebral Hemorrhage” PMID: 26022637) or primary source not cited (e.g. references 26 and 27 are not the primary studies that support PRx as being more reliable than ICP threshold alone in predicting death) in several places. Note the examples given are not the only missing/inappropriate references identified.
  • Major ethical concern: it is not stated whether patients (or next of kin) were fully informed and gave consent for inclusion in this study.
  • The discussion contains a large amount of extraneous details that obscure the key discussion points. I strongly advise that the authors to refine their discussion.

Minor comments:

Introduction

  • Requires more detail on the importance of each of the physiological parameters in HICH.

Methods

  • Not clear how mean data values are calculated. Method section states “measurement intervals for all data were 3 seconds and the were calculated within 1 hour as the time segment”, yet in the results only the mean (one assumes for the entire 3-day data collection window) are given.

Results

  • Table 1: Gender “Current result” missing the ‘chi’ symbol and missing definition of acronyms at bottom of table.
  • Figure 1: “sound prognosis” should be changed to “good prognosis” to remain consistent with the rest of the manuscript. Significance bar missing on figure.
  • Figure 2: does not provide any insight into the data. Remove.

Discussion

  • Line 141: “The HICH induced…”; there are other more appreciable contributors to HICH pathophysiology including vasospasm and altered cerebral hemodynamics. These are the key events that decompressive surgery aims to prevent. Would replace text that discusses compressive displacement and peripheral immune infiltration.
  • Line 156 “The sharp increase in ICP…”; false. Remove.
  • Highlight that the majority of HICH treatment approaches are drawn from traumatic brain injury research.
  • Line 174 “These results were contradictory…”; expand on Vik’s findings and potential reasons why.
  • Line 178 “Furthermore, the lack of…”; sentence does not make sense.
  • Line 181 “Analysis of the data…”; not clear whose data is being discussed. Is it published data or the author’s own data? If the author’s own, this data has not been included in the manuscript. If published data, it has not been cited.
  • Line 200 “At normal cerebrovascular…”; consider rewording this sentence. Sounds like MAP drops to directly alter ICP when it is more of a feedback loop than a direct interaction.
  • Line 203 “synthetic reaction correlation” not clear what this means.
  • Paragraph at line 255 should be in the introduction.
  • Line 243 “…reserve of the cerebrospinal…” appears to be words missing at the end of the sentence.

Author Response

Major:

Q1:Data on hematoma volume (pre- and post-operative) and location needs to be included. Hematoma volume and location is a powerful prognostic and is a confounding variable that needs to be accounted for.Data on baseline blood pressure not given therefor not possible to determine whether patients are hypertensive.

Response:Thank you for your opinion. All patients suffer from hypertension. Intravenous blood pressure lowering drugs have been used to control blood pressure in the emergency department. We can't obtain relevant data, but the blood pressure of patients is roughly controlled within the normal range during treatment in our department. At the same time, in order to avoid the difference in prognosis caused by differences in different functional areas.All patients selected in the study were basal ganglia hematomas, oval in shape, with preoperative hematoma volume of 20-50ml , hematoma volume in the good prognosis group (35.41±6.63ml), hematoma volume in the poor prognosis group (37.38±6.40ml), there was no significant difference, postoperative hematoma clearance was satisfactory, and all patients had no rebleeding during hospitalization.

Q2:This study does not inform better treatment of HICH. Use of CPPopt in HICH has been previously reported (“Optimal cerebral perfusion pressure in patients with intracerebral hemorrhage: an observational case series” PMID: 24666981). Would advise that the authors adjust their manuscript to focus on prognostication.

Response:Thank you for your comments.For the new treatment plan for hich, our work has really made no substantive progress. We will continue to study hard. Please give us a chance.We apologize for the misunderstanding caused by the content distribution of our article. The purpose of our research includes treatment, because we will adjust the treatment plan according to ICP related parameters during the treatment process, such as dehydration, the timing of CT Reexamination, and even whether we need reoperation. We don't mention much in the article, which is our mistake. We have recognized this deficiency and have revised the content of the article.

Q3:References missing (e.g. Lines 45-47 refer to “Guidelines for the Management of Spontaneous Intracerebral Hemorrhage” PMID: 26022637) or primary source not cited (e.g. references 26 and 27 are not the primary studies that support PRx as being more reliable than ICP threshold alone in predicting death) in several places. Note the examples given are not the only missing/inappropriate references identified.

Response:Thank you for your suggestions. We have added relevant references as follows: Hemphill JC 3rd, Greenberg SM, Anderson CS, Becker K, Bendok BR, Cushman M, et al. (2015). Guidelines for the management of spontaneous intracerebral hemorrhage. Stroke, 46(7), 2032-2060. doi: 10.1161/STR.0000000000000069.

References 26 and 27 mentioned that PRX is more reliable than ICP threshold in predicting death, but we didn't consider that this article is not mainly about this problem. Our initial meaning is to prove that this view is recognized by many scholars with more articles. I hope the professor can accept our approach.

Q4:Major ethical concern: it is not stated whether patients (or next of kin) were fully informed and gave consent for inclusion in this study.

Response:Thank you for your suggestions. It is the omission of our article.In fact,all patients were in coma, and their families signed the informed consent for operation and ICP probe.

Q5:The discussion contains a large amount of extraneous details that obscure the key discussion points. I strongly advise that the authors to refine their discussion.

Response:Thank you for your suggestion. Several parameters are mentioned in the manuscript. The discussion part of the article focuses on analyzing the significance of each parameter, and some contents are added to explain the relationship between different parameters and the initial idea of studying these parameters. We have made some changes in the article. I hope professor can understand and thank you for giving us this opportunity.

Minor:

Q1:Introduction:

Requires more detail on the importance of each of the physiological parameters in HICH.

Response:Thank you for your suggestion. We have revised the manuscript and briefly introduced the significance of he physiological parameters in the introduction. Because there are many details, we will introduce them in detail in the discussion.

Q2:Methods:

Not clear how mean data values are calculated. Method section states “measurement intervals for all data were 3 seconds and the were calculated within 1 hour as the time segment”, yet in the results only the mean (one assumes for the entire 3-day data collection window) are given.

Response:Thank you for your comments. The default setting of our data collection system is to collect one data every 3S as the original data, and then we process the original data twice, taking one hour as the unit to get the average value mentioned in the article.

Q3:Results:

Table 1: Gender “Current result” missing the ‘chi’ symbol and missing definition of acronyms at bottom of table.

Response:Thank you for your advice. All abbreviations are introduced at the end of the article.

Figure 1: “sound prognosis” should be changed to “good prognosis” to remain consistent with the rest of the manuscript. Significance bar missing on figure.Figure 2: does not provide any insight into the data. Remove.

Response:Thank you for your comments. Figures 1 and Figures 2 have been modified in the article.

Q4:Discussion:

Line 141: “The HICH induced…”; there are other more appreciable contributors to HICH pathophysiology including vasospasm and altered cerebral hemodynamics. These are the key events that decompressive surgery aims to prevent. Would replace text that discusses compressive displacement and peripheral immune infiltration.

Response:Thank you for your suggestions. My understanding is as follows: decompression surgery can increase brain space compensation ability, alleviate brain tissue displacement and avoid further brain injury. However, this sentence generally explains the pathophysiological mechanism of the edema zone around the hematoma. At the same time, our department relies more on TCD for the monitoring of vasospasm, so it is not explained in detail. The changes of cerebral hemodynamics are mentioned in the text through the monitoring parameters.

Line 156 “The sharp increase in ICP…”; false. Remove.

Response:Thank you for your suggestion. We have deleted it.

Highlight that the majority of HICH treatment approaches are drawn from traumatic brain injury research.

Response:At present, most of the research on ICP parameters draws lessons from craniocerebral trauma. The initial definition of ICP threshold in China's hich guidelines is also based on craniocerebral trauma. At the same time, they have something in common with most treatment schemes, so we explain it in this way. I hope that you can put forward valuable opinions. Thank you!

Line 174 “These results were contradictory…”; expand on Vik’s findings and potential reasons why.

Response:Thank you for your criticism. What we wrote means that the results are different from Vik's findings, indicating that the results may be different in other large sample data.

Line 178 “Furthermore, the lack of…”; sentence does not make sense.

Response:Thank you for your advice. We have deleted this sentence.

Line 181 “Analysis of the data…”; not clear whose data is being discussed. Is it published data or the author’s own data? If the author’s own, this data has not been included in the manuscript. If published data, it has not been cited.

Response:Thank you for your advice. We found this phenomenon in a very few cases, so we thought about the possible causes, and then thought of the parameter DICP20 to further explain the significance of existence. The purpose of this sentence is also to explain why we should study DICP20. We have revised this sentence and hope that you can forgive the misunderstanding caused.

Line 200 “At normal cerebrovascular…”; consider rewording this sentence. Sounds like MAP drops to directly alter ICP when it is more of a feedback loop than a direct interaction.

Response:Thank you for your comments. We have revised the sentence. What we want to express is that there is a certain coefficient relationship between MAP and ICP.

Line 203 “synthetic reaction correlation” not clear what this means.

Response:Thank you for your comments. We are deeply sorry for the inaccuracy of our expression.We have revised the sentence.

Paragraph at line 255 should be in the introduction.

Response:Thank you for your comments. The introduction of RAP has been mentioned in the introduction, so we deleted this sentence.

Line 243 “…reserve of the cerebrospinal…” appears to be words missing at the end of the sentence.

Response:Thank you for your advice. We are very sorry for this mistake. We have corrected this sentence.